# Generative Modeling of Molecular Dynamics Trajectories

**Bowen Jing** [* 1]   **Hannes Stark** [* 1]   **Tommi Jaakkola** [1]   **Bonnie Berger** [1 2]

## Abstract

Molecular dynamics (MD) is a powerful technique for studying microscopic phenomena, but its computational cost has driven significant interest in the development of deep learning-based surrogate models. We introduce generative modeling of molecular trajectories as a paradigm for learning flexible multi-task surrogate models of MD from data. By conditioning on appropriately chosen frames of the trajectory, we show such generative models can be adapted to diverse tasks such as forward simulation, transition path sampling, and trajectory upsampling. By alternatively conditioning on part of the molecular system and inpainting the rest, we also demonstrate the first steps towards dynamics-conditioned molecular design. We validate the full set of these capabilities on tetrapeptide simulations and show that our model can produce reasonable ensembles of protein monomers. Altogether, our work illustrates how generative modeling can unlock value from MD data towards diverse downstream tasks that are not straightforward to address with existing methods or even MD itself.

## 1. Introduction

Numerical integration of Newton's equations of motion at atomic scales, known as *molecular dynamics* (MD), is a widely-used technique for studying diverse molecular phenomena in chemistry, biology, and other molecular sciences (Alder & Wainwright, 1959; Rahman, 1964; Verlet, 1967; McCammon et al., 1977). While general and versatile, MD is computationally demanding due to the large separation in timescales between integration steps and relevant molecular phenomena. Thus, a vast body of literature spanning several decades aims to accelerate or enhance the sampling efficiency of MD simulation algorithms (Ryckaert et al.,

1977; Darden et al., 1993; Sugita & Okamoto, 1999; Laio & Parrinello, 2002; Anderson et al., 2008; Shaw et al., 2009). More recently, learning surrogate models of MD has become an active area of research in deep generative modeling (Noé et al., 2019; Zheng et al., 2023; Klein et al., 2024; Schreiner et al., 2024; Jing et al., 2024). However, existing training paradigms fail to fully leverage the rich dynamical information in MD training data, restricting their applicability to a limited set of downstream problems.

In this work, we propose MDGEN, a novel paradigm for fast, general-purpose surrogate modeling of MD based on *direct generative modeling of simulated trajectories*. Different from previous works, which learn the autoregressive transition density or equilibrium distribution of MD, we formulate end-to-end generative modeling of full trajectories viewed as *time-series* of 3D molecular structures. Akin to how image generative models were extended to videos (Ho et al., 2022), our framing of the problem augments single-structure generative models with an additional time dimension, opening the door to a larger set of forward and inverse problems to which our model can be applied. When provided (and conditioned on) the initial "frame" of a given system, such generative models serve as familiar surrogate forward simulators of the reference dynamics. However, by providing other kinds of conditioning, these "molecular video" generative models also enable highly flexible applications to a variety of inverse problems not possible with existing surrogate models. In sum, we formulate and showcase the following novel capabilities of MDGEN:

- *Forward simulation*—given the initial frame of a trajectory, we sample a potential time evolution.

- *Interpolation*—given the frames at the *two endpoints* of a trajectory, we sample a plausible path connecting the two. In chemistry, this is known as *transition path sampling* and is important for studying reactions.

- *Upsampling*—given a trajectory with timestep $\Delta t$ between frames, we upsample the "framerate" to $\Delta t/M$.

- *Inpainting*—given part of a molecule and its trajectory, we *generate the rest of the molecule* (and its time evolution) to be consistent with the known part of the trajectory. This ability could be applied to design molecules to scaffold desired dynamics.

*Equal contribution [1]CSAIL, Massachusetts Institute of Technology [2]Dept. of Mathematics, Massachusetts Institute of Technology. Correspondence to: Bowen Jing <bjing@mit.edu>, Hannes Stark <hstark@mit.edu>.

*Accepted at the 1st Machine Learning for Life and Material Sciences Workshop at ICML 2024.* Copyright 2024 by the author(s).

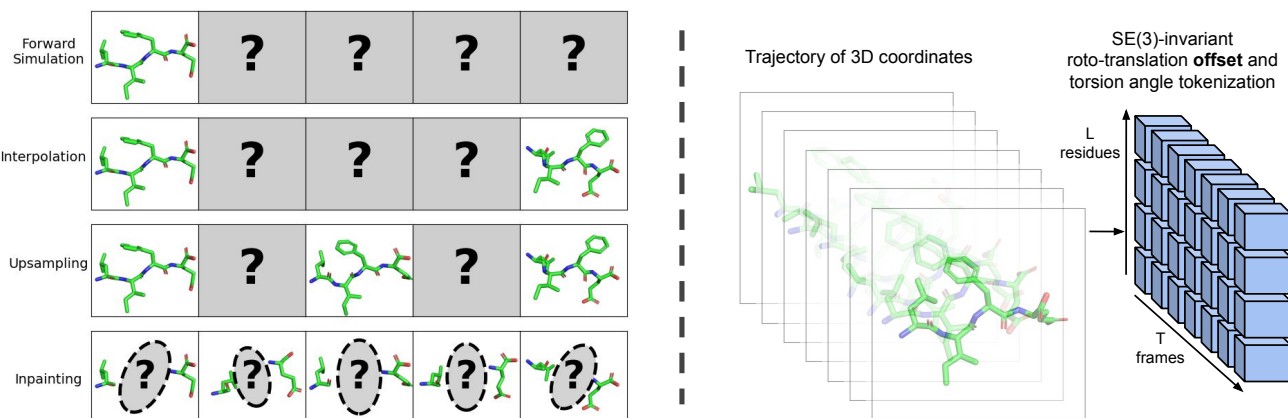

Figure 1. (*Left*) Tasks: generative modeling of MD trajectories addresses several tasks by conditioning on different parts of a trajectory. (*Right*) Method: We tokenize trajectories of $T$ frames and $L$ residues into an $(T \times L)$-array of SE(3)-invariant tokens encoding roto-translation offsets from *key frames* and torsion angles. Using *stochastic interpolants*, we generate such tokens from Gaussian noise.

These tasks are conceptually illustrated in Figure 1. While the forward simulation task aligns with the typical modeling paradigm of approximating the data-generating process, the others represent novel capabilities on scientifically important inverse problems *not straightforward to address even with MD itself*. As such, our framework presents a new perspective on how to unlock value from MD simulation with machine learning towards diverse downstream objectives. We highlight further exciting possibilities opened up by our framework in Section 4.

We evaluate MDGEN on the forward simulation, interpolation, upsampling, and inpainting tasks on tetrapeptides in a *transferable* setting (i.e., generalizing to test peptides). Our model accurately reproduces free energy surfaces and dynamical content such as torsional relaxation and Markov state fluxes, provides realistic transition paths between arbitrary pairs of metastable states, and recovers fast dynamical phenomena below the sampling threshold of coarse-timestep trajectories. In preliminary steps toward dynamics-scaffolded design, we show that molecular inpainting with MDGEN obtains much higher sequence recovery than inverse folding methods based on one or two static frames. Finally, we evaluate MDGEN on forward simulation of protein monomers and find that its ensembles' statistical properties surpass the fidelity of ensembles from multiple sequence alignment (MSA) subsampling with AlphaFold (Del Alamo et al., 2022).

## 2. Method

### 2.1. Tokenizing Molecular Trajectories

For further background on MD and Stochastic Interpolants, please see Appendix A. Given a chemical specification of a molecular system with $N$ atoms, our aim is to learn a generative model over time-series $\boldsymbol{\chi} \equiv [\mathbf{X}_1, \dots \mathbf{X}_T]$ of cor-

responding molecular structures $\mathbf{X}_i \in \mathbb{R}^{3N}$ for some trajectory length $T$. In this work, we specialize to MD trajectories of short peptides (Sections 3.1–D.1) or single-chain proteins (D.1); thus, our chemical specifications are amino acid sequences $A = \{1 \dots 20\}^L$. We adopt a $SE(3)$-based parameterization of molecular structures (Jumper et al., 2021; Yim et al., 2023), such that the all-atom coordinates of each amino acid residue are described by a roto-translation (i.e., element of $SE(3)$) and seven torsion angles $\boldsymbol{\chi}_t^l = ((R, \mathbf{t}), (\psi, \phi, \omega, \chi_1 \dots \chi_4)), \quad \boldsymbol{\chi} \in \left( [SE(3) \times \mathbb{T}^7]^L \right)^T$ where sub-/superscripts indicate time/residue indices.

To learn a generative model over this space of roto-translations and torsion angles, one natural choice is to employ diffusion or flow-based models for protein structures explicitly designed for such representations (Yim et al., 2023; Lin & AlQuraishi, 2023). However, these architectures rely on expensive edge-based and IPA-based updates, which can be memory-intensive to replicate across a large number of trajectory frames. Instead, we leverage the fact that we are concerned with *conditional trajectory generation*—meaning that there always exists at least one frame in the trajectory with un-noised roto-translations which we do not need to generate and can reference in the modeling process. Inspired by analogy to video compression, we refer to such frames as *key frames* and parameterize the roto-translations of remaining structures as *offsets relative to the key frames*. Specifically, given $K$ key frames $i_1 \dots i_K$ we tokenize residue $j$ in frame $t$ as:

$$\boldsymbol{\chi}_t^j = \left( g_{i_1}^{-1} g_j, \dots, g_{i_K}^{-1} g_j, \boldsymbol{\tau}_j \right) \in (\hat{\mathbb{Q}}^+ \oplus \mathbb{R}^3)^K \times (S^2)^7 \subset \mathbb{R}^{7K+14} \quad (1)$$

where $g_j \in SE(3)$ represents the roto-translation and $\boldsymbol{\tau}_j$ the torsion angles of residue $j$. Namely, we convert the relative roto-translational offsets $g_i^{-1} g_j$ to unit quaternions with positive real part $\hat{\mathbb{Q}}^+ \subset \mathbb{R}^4$ (representing the rota-

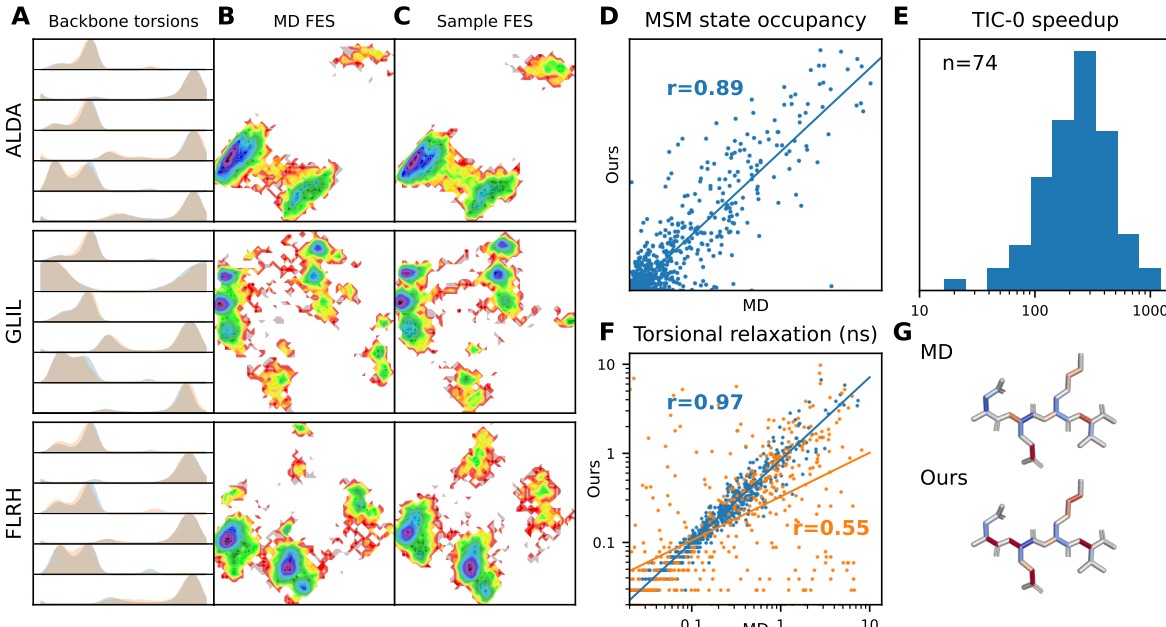

*Figure 2.* **Forward simulation evaluations on test peptides.** (**A**) Torsion angle distributions for the six backbone torsion angles from MD trajectories (orange) and sampled trajectories (blue). (**B, C**) Free energy surfaces along the top two TICA components computed from backbone and sidechain torsion angles. (**D**) Markov State Model occupancies computed from MD trajectories versus sampled trajectories, pooled across all test peptides ($n = 1000$ states total). (**E**) Wall-clock decorrelation times of the first TICA component under MD versus our model rollouts. (**F**) Relaxation times of all torsion angles, pooled across all test peptides (508 backbone and 722 sidechain torsions in total) computed from MD versus sampled trajectories. (**G**) Torsion angles in the tetrapeptide `AAAA` colored by the decorrelation time computed from MD (top) and from rollout trajectories (bottom).

tions) and translation vectors in $\mathbb{R}^3$, and convert torsion angles to points on the unit circle, obtaining a $(7K + 14)$-dimensional token for each residue in every frame. To *untokenize* to an all-atom frame $\mathbf{X}_i \in \mathbb{R}^{3N}$, we read off the torsion angles from the unit circle and apply the key frame roto-translations $g_{i_k}$ to the generated offsets to obtain absolute roto-translations $\hat{g}_j$. The offsets and torsion angles are $SE(3)$-invariant; thus, we obtain a representation of molecular trajectories as an $(T \times L)$-array of $SE(3)$-invariant tokens: $\chi \in \mathbb{R}^{T \times L \times (7K+14)}$. This representation allows us to directly apply a flow model architecture to generate the tokens (details in Appendix B.1.

### 2.2. Conditional Generation

We present the precise specifications of conditioning settings in Table 2. Depending on the task, we choose the key frames to be the first frame $g_1$ or the first and last frames $g_1, g_T$. Each task is further characterized by providing the ground-truth tokens of known frames or residues as additional inputs to the model. Meanwhile, mask tokens are provided for the unknown frames and residues that the model generates. For example, in the upsampling setting, we provide ground-truth tokens every $M$ frames, while mask tokens are provided for all other frames. We note that in the inpainting setting, the model accesses the roto-translations $g$ of designed residues at the trajectory endpoints via the key frames, constituting a slight departure from the full inpainting setting.

*Table 1.* JSD between sampled and ground-truth distributions, with replicate MD as baselines. 100 ns represents oracle performance.

| C.V. | Ours | 10 ns | 1 ns | 100 ps | 100 ns |
|---|---|---|---|---|---|
| Torsions (bb) | **.130** | .145 | .212 | .311 | .103 |
| Torsions (sc) | **.093** | .111 | .261 | .403 | .055 |
| Torsions (all) | **.109** | .125 | .240 | .364 | .076 |
| TICA-0 | **.230** | .323 | .432 | .477 | .201 |
| TICA-0,1 joint | **.316** | .424 | .568 | .643 | .268 |
| MSM states | **.235** | .363 | .493 | .527 | .208 |
| Runtime | 60s | 1067s | 107s | 11s | 3h |

## 3. Experiments

We focus on *tetrapeptides* as our main molecule class for evaluation. Additional tasks are in Appendix 3.3—namely, inpainting for dynamics-conditioned design, long trajectories with Hyena (Poli et al., 2023), and scaling to simulations of protein monomers. Separate models are trained for each experimental setting. To obtain tetrapeptide MD trajectories for training and evaluation, we run MD for ≈3000 training, 100 validation, and 100 test tetrapeptides for 100 ns. For proteins, we use the ATLAS dataset (Vander Meersche et al., 2024), which 300 ns of MD for each of 1390 proteins. Unless otherwise specified, models are trained with trajectory timesteps of $\Delta t = 10$ ps. Our default baselines consist of *replicate MD simulations* ranging from 10 ps to 100 ns. Our experiments make extensive use of Markov State Models (MSMs), which are a popular and tested coarse-grained

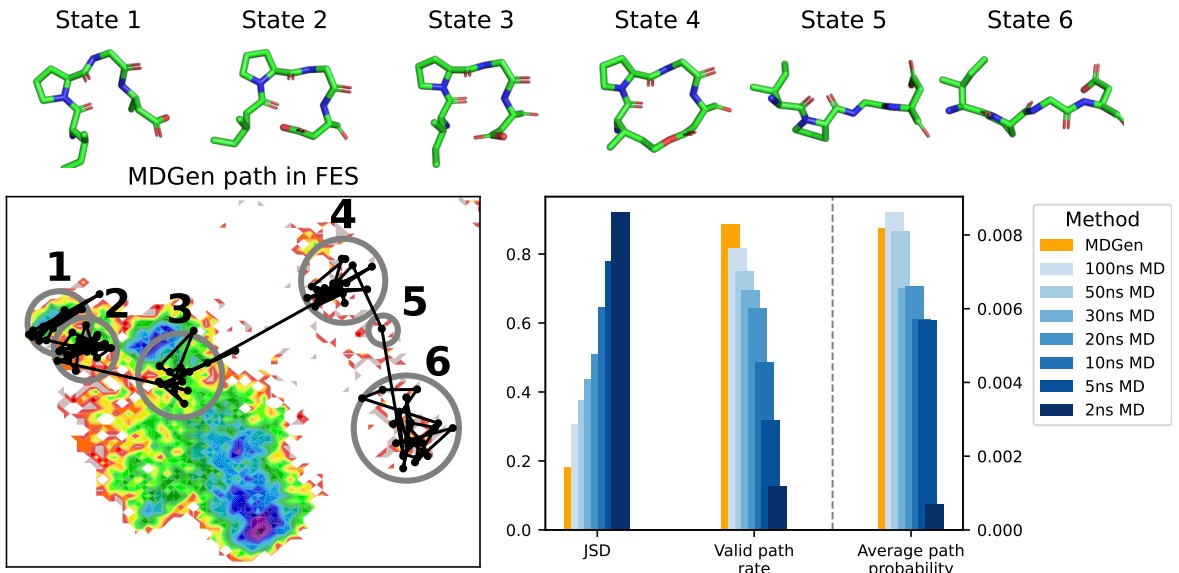

*Figure 3.* **Transition path sampling results.** (**Top**) Intermediate states of one of the 1-nanosecond interpolated trajectories between two metastable states for the test peptide `IPGD`. (**Bottom Left**) The corresponding trajectory on the 2D free energy surface of the top two TICA components (more examples in Figure 9). (**Bottom Right**) Statistics averaged over 100 test peptides and 1000 paths for each of them. Shown are JSD, fraction of drawn paths that are valid transition paths, and average path likelihood of our discretized transitions under the reference MSM compared to discrete transitions drawn from the reference MSM or alternative MSMs built from replica simulations of varying lengths.

representation of MD (Prinz et al., 2011; Noé et al., 2013). Appendix C provides further details on MSMs with more results in Appendix D.

### 3.1. Forward Simulation

**Distributional similarity.** We train a model to sample 10 ns trajectories conditioned on the first frame. By chaining together successive model rollouts at inference time, we obtain 100 ns trajectories for each peptide to compare with ground-truth simulations. We report the Jensen-Shannon divergence (JSD) between the ground-truth and emulated trajectories along various collective variables shown in Figure 2 and Table 2. The first set of these are the individual torsion angles (backbone and sidechains) in each tetrapeptide. The second set of variables are the top independent components obtained from *time-lagged independent components analysis* (TICA), representing the slowest dynamic modes of the peptide. By each of these collective variables, MDGEN demonstrates excellent distributional similarity to the ground truth, approaching the accuracy of replicate 100-ns simulations. To more stringently assess the ability to locate and populate modes in the joint distribution over state space, we build Markov State Models (MSMs) for each test peptide using the MD trajectory, extract the corresponding metastable states, and compare the ground-truth and emulated distributions over metastable states. Our model captures the relative ranking of states (Figure 2D).

**Dynamical content.** We compute the dynamical properties of each tetrapeptide in terms of the *decorrelation time* of

each torsion angle from the MD simulation and from our sampled trajectory. Intuitively, this assesses if our model can discriminate between slow- and fast-relaxing torsional barriers. The correlation between true and predicted relaxation timescales is plotted in Figure 2F, showing excellent agreement for sidechain torsions. To assess coarser but higher-dimensional dynamical content, we compute the *flux matrix* between all pairs of distinct metastable states using ground-truth and sampled trajectories and find substantial Spearman correlation between their entries (mean $\rho = 0.67 \pm 0.01$; Figure 8).

**Sampling speed.** Averaged across test peptides, our model samples 100 ns-equivalent trajectories in $\approx$60 GPU-seconds, compared to $\approx$3 GPU-hours for MD. To quantify the speedup more rigorously, we compute the decorrelation wall-clock times along the slowest independent component from TICA, capturing how quickly the simulation traverses the highest barriers in state space. These times are plotted in Figure 2E, showing that our model achieves an speedup of 10x–1000x over the MD simulation for 78 out of 100 peptides (the other 22 peptides did not fully decorrelate).

### 3.2. Interpolation

In the *interpolation* or *transition path sampling* setting, we train a model to sample 1 ns trajectories conditioned on the first and last frames. For evaluation, we identify the two most well-separated states (i.e., with the least flux between them) for each test peptide and sample an ensemble of 1000 transition paths between them. Figure 3 shows an

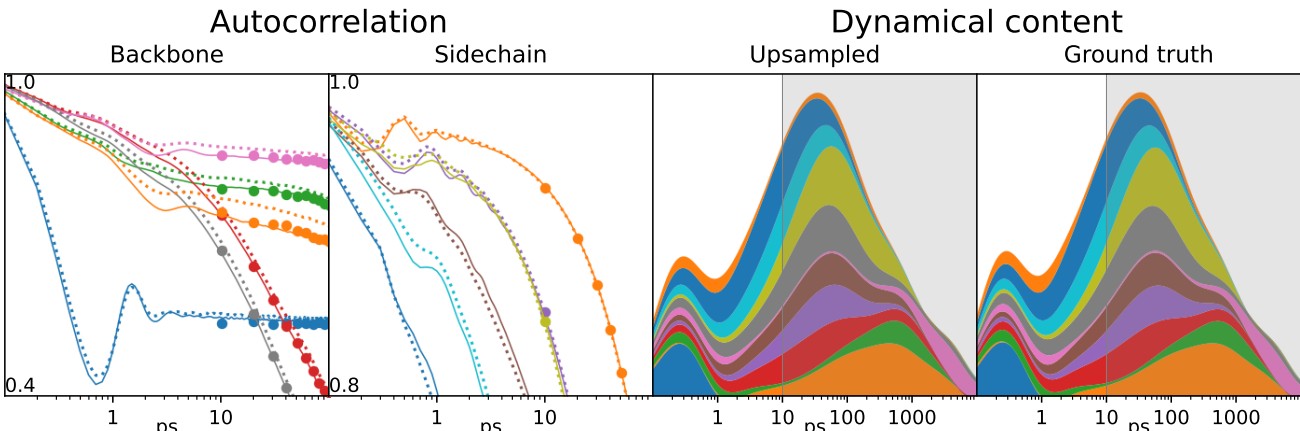

*Figure 4.* **Recovery of fast dynamics via trajectory upsampling** for peptide `GTLM`. (*Left*) Autocorrelations of each torsion angle from (—) the original 100 fs-timestep trajectory, (●) the subsampled 10 ns-timestep trajectory, and (···) the reconstructed 100 fs-timestep trajectory (all length 100 ns). (*Right*) Dynamical content as a function of timescale from the upsampled vs. ground truth trajectories, stacked for all torsion angles (same color scheme). The subsampled trajectory contains only the shaded region and our model recovers the unshaded region. Further examples in Figure 10.

example of such a sampled path, which passes through several intermediate states on the free energy surface.

To evaluate the accuracy of these sampled transitions, we cannot directly compare with MD trajectories since, in most cases, there are zero or very few 1-ns transitions between the two selected states (by design, the transition is a *rare event*). Thus, we instead discretize the trajectory over MSM metastable states and evaluate the *path likelihood* under the transition path distribution from the reference MSM (details in Appendix C.3). We also report the fraction of non-zero probability paths and the JSD between the distribution of visited states from our path distribution versus the transition path distribution of the reference MSM. For baselines, we sample transition paths from MSMs constructed from replicate MD simulations of varying lengths and compute the metrics for their path ensembles under the reference MSM.

As shown in Figure 3, our paths have higher likelihoods than those sampled from any replicate MD MSM shorter than 100ns, which is the length of the reference MD simulation itself. Moreover, MDGEN's ensembles have the best JSDs to the distribution of visited states of the reference MD MSM and the highest fraction on valid non-zero probability paths. Hence, our model enables zero-shot sampling of trajectories corresponding to arbitrary rare transitions.

### 3.3. Upsampling

In the *upsampling* setting, we train MDGEN to upsample trajectories saved with timestep 10 ps to a finer timestep of 100 fs, representing a 100x upsampling factor. To evaluate if the upsampled trajectories accurately capture the fastest dynamics, we compute the *autocorrelation function* $\langle \cos(\theta_t - \theta_{t+\Delta t}) \rangle$ of each torsion angle in the test peptides as a function of lag time $\Delta t$ ranging from 100 fs to 100 ps.

Representative examples of ground truth, subsampled, and reconstructed autocorrelation functions for two test peptides are shown in Figure 4 (further examples in Figure 10). We further compute the *dynamical content* as the negative derivative of the autocorrelation with respect to log-timescale, which captures the extent of dynamic relaxations occuring at that timescale (Shaw et al., 2009). These visualizations highlight the significant dynamical information absent from the subsampled trajectory and which are accurately recovered by our model. In particular, our model distinctly recovers the *oscillations* of certain torsion angles as seen in the non-monotonicity of the autocorrelation function at sub-picosecond timescales; these features are completely missed at the original sampling frequency.

## 4. Discussion

**Opportunities.** Similar to the foundational role of video generative models for understanding the macroscopic world (Yang et al., 2024), MD trajectory generation could serve as a multitask, unifying paradigm for deep learning over the microscopic world. Interpolation can be more broadly framed as *hypothesis generation* for mechanisms of arbitrary molecular phenomena, especially when only partial information about the end states is supplied. Molecular inpainting could be a general technique to design molecular machinery by scaffolding more fine-grained and complex dynamics, for example redesigning proteins to enhance rare transitions observed only once in a simulation or *de novo* design of enzymatic mechanisms and motifs. Other types of conditioning not explored in this work may lead to further applications, such as conditioning over textual or experimental descriptors of the trajectory. Future availability of more ground truth MD trajectory data for diverse chemical systems could be a chief enabler of such work.

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

# A. Background

**Molecular dynamics.** At a high level, the aim of molecular dynamics is the integrate the equations of motion $M_i \ddot{\mathbf{x}}_i = -\nabla_{\mathbf{x}_i} U(\mathbf{x}_1 \dots \mathbf{x}_N)$ for each particle $i$ in a molecular configuration $(\mathbf{x}_1 \dots \mathbf{x}_N) \in \mathbb{R}^{3N}$, where $M_i$ is the mass and $U$ is the potential energy function (or *force field*) $U : \mathbb{R}^{3N} \to \mathbb{R}$. However, these equations of motion are often modified to include a *thermostat* in order to model contact with surroundings at a given temperature. For example, the widely-used *Langevin thermostat* transforms the equations of motion into a stochastic diffusion process of the equations $d\mathbf{x}_i = \mathbf{p}_i/M_i \, dt$ and $d\mathbf{p}_i = -\nabla_{\mathbf{x}_i} U \, dt - \gamma \mathbf{p}_i \, dt + \sqrt{2M_i \gamma kT} \, d\mathbf{w}$, where $\mathbf{p}_i$ are the momenta. By design, this process converges to the *Boltzmann distribution* of the system $p(\mathbf{x}_1 \dots \mathbf{x}_N) \propto e^{-U/kT}$. To incorporate interactions with solvent molecules—ubiquitous in biochemistry—one includes a box of surrounding solvent molecules as part of the molecular system (explicit solvent) or modifies the force field $U$ to model their effects (implicit solvent). In either case, only the positions $\mathbf{x}_i$ of non-solvent atoms are of interest, and their time evolution constitutes (for our purposes) the *MD trajectory*. Thus, MD can be mathematically described as a partially observed Markovian diffusion process.

**Deep learning for MD.** An emerging body of work seeks to approximate the distributions over configurations $\mathbf{X} = (\mathbf{x}_1 \dots \mathbf{x}_N)$ arising from MD with deep generative models. Fu et al. (2023), Timewarp (Klein et al., 2024), and ITO (Schreiner et al., 2024) learn the *transition density* $p(\mathbf{X}_{t+\Delta t} \mid \mathbf{X}_t)$ and emulate MD trajectories via simulation rollouts of the learned model. On the other hand, *Boltzmann generators* (Noé et al., 2019; Köhler et al., 2021; Garcia Satorras et al., 2021; Midgley et al., 2022; 2024) directly approximate the stationary Boltzmann distribution, forgoing any explicit modeling of dynamics. In particular, Boltzmann-targeting diffusion models trained with frames from MD trajectories have demonstrated promising scalability and generalization to protein systems (Zheng et al., 2023; Jing et al., 2024). However, these works have focused exclusively on forward simulation and have not explored joint modeling of entire trajectories $(\mathbf{X}_t \dots \mathbf{X}_{t+N\Delta t})$ or the inverse problems accessible under such a formulation.

**Stochastic interpolants.** We build our MD trajectory generative model under the *stochastic interpolants* framework: Given a continuous distribution $p_1 \equiv p_{\text{data}}$ over $\mathbb{R}^n$, stochastic interpolants (Albergo & Vanden-Eijnden, 2022; Albergo et al., 2023; Lipman et al., 2022; Liu et al., 2022), provide a method for learning continuous flow-based models $d\mathbf{x} = v_\theta(\mathbf{x}, t) \, dt$ transporting a prior distribution $p_0$ (e.g., $p_0 \equiv \mathcal{N}(0, \mathbf{I})$) to the data $p_1$. To do so, one defines intermediate distributions $\mathbf{x}_t \sim p_t, t \in (0, 1)$ via $\mathbf{x}_t = \alpha_t \mathbf{x}_1 + \sigma_t \mathbf{x}_0$ where $\mathbf{x}_0 \sim p_0$ and $\mathbf{x}_1 \sim p_1$ and the interpolation path satisfies $\alpha_0 = \sigma_1 = 0$ and $\alpha_1 = \sigma_0 = 1$. A neural network $v_\theta : \mathbb{R}^n \times [0, 1] \to \mathbb{R}^n$ is trained to approximate the time-evolving flow field $v_\theta(\mathbf{x}_t, t) \approx v(\mathbf{x}_t, t) \equiv \mathbb{E}_{\mathbf{x}_0, \mathbf{x}_1 | \mathbf{x}_t}[\dot{\alpha}_t \mathbf{x}_1 + \dot{\sigma}_t \mathbf{x}_0]$ which satisfies the transport equation $\partial p_t / \partial t + \nabla \cdot (p_t v_t) = 0$. Hence, at convergence, noisy samples $\mathbf{x}_0 \sim p_0$ can be evolved under $v_\theta$ to obtain data samples $\mathbf{x}_1 \sim p_1$. When parameterized with transformers (Vaswani et al., 2017), stochastic interpolants are state-of-the-art in image generation (Esser et al., 2024). In particular, we adopt the notation, architecture, and training framework of Scalable Interpolant Transformer (SiT) (Ma et al., 2024), to which we refer for further exposition.

# B. Method Details

## B.1. Flow Model Architecture

Our base modeling task is to generate a distribution over $\mathbb{R}^{T \times L \times (7K+14)}$ conditioned on roto-translations of one or more key frames $g_{i_1} \dots g_{i_K}$, and (in most settings) amino acid identities $A$. To do so, we learn a flow-based model via the stochastic interpolant framework described in SiT (Ma et al., 2024) and parameterize a velocity network $v_\theta(\cdot \mid g_{i_1} \dots g_{i_K}, A) : \mathbb{R}^{T \times L \times (7K+14)} \times [0, 1] \to \mathbb{R}^{T \times L \times (7K+14)}$. To condition on the key frames and amino acids, we first provide the sequence embedding to several IPA layers (Jumper et al., 2021) that embed the key frame roto-translations; these conditioning representations (which are $SE(3)$-invariant) are broadcast across the time axis and added to the input embeddings. The main trunk of the network consists of alternating attention blocks across the residue index and across time, with the construction of each block closely resembling DiT (Peebles & Xie, 2023). Sidechain torsions and roto-translation offsets, when available, are directly provided to the model as conditioning tokens.

In the molecular inpainting setting where we also *generate* the amino acid identities, we additionally require a generative framework over these discrete variables. While several formulations of discrete diffusion or flow-matching are available (Hoogeboom et al., 2021; Austin et al., 2021; Campbell et al., 2022; 2024), we select Dirichlet flow matching (Stark et al., 2024) as it is most compatible with the continuous-space, continuous-time stochastic interpolant framework used for the positions. Specifically, we place the amino acid identities on the 20-dimensional probability simplex (one per amino acid),

augment the token representations with these variables, and regress against a $T \times L \times (7K + 14 + 20)$-dimensional vector field. Further details in Appendix B.2.

*Table 2.* Conditional generation settings. $g$: roto-translations, $\boldsymbol{\tau}$: torsions, $A$: residue identities $M$: upsampling factor. Superscripts indicate residue index and subscripts indicate frame index. For inpainting, we find excluding identities and torsions to reduce overfitting.

| Setting | Key frames | Generate | Conditioned on | Token dim. |
|---|---|---|---|---|
| Forward simulation | $g_1$ | $g_{1\cdots T}, \boldsymbol{\tau}_{1\cdots T}$ | $g_1, \boldsymbol{\tau}_1, A$ | 21 |
| Interpolation | $g_1, g_T$ | $g_{1\cdots T}, \boldsymbol{\tau}_{1\cdots T}$ | $g_{1,T}, \boldsymbol{\tau}_{1,T}, A$ | 28 |
| Upsampling | $g_1$ | $g_{1\cdots T}, \boldsymbol{\tau}_{1\cdots T}$ | $g_{1+\{1,2,\cdots\}M}, \boldsymbol{\tau}_{1+\{1,2,\cdots\}M}, A$ | 21 |
| Inpainting | $g_1, g_T$ | $g_{1\cdots T}, A$ | $g^{\text{known}}_{1\cdots T}$ | 7 (+20) |

**Notation.** Here, as in the main text, we use the following notation:

- $T$: number of trajectory frames

- $L$: number of amino acids

- $K$: number of key frames, with indicies $i_1 \dots i_K$

In Algorithms 1–3 below, we modify the architectures of DiffusionTransformerAttentionLayer and DiffusionTransformerFinalLayer from DiT (Peebles & Xie, 2023). Elements from these layers are also then incorporated into our custom InvariantPointAttentionLayer.

---

**Algorithm 1** Velocity network

---

**Input:** noisy tokens $\boldsymbol{\chi} \in \mathbb{R}^{T \times L \times (7K+14)}$, conditioning tokens $\boldsymbol{\chi}_{\text{cond}} \in \mathbb{R}^{T \times L \times (7K+14)}$, key frame roto-translations $g_{i_1} \dots g_{i_K} \in \left(SE(3)^L\right)^K$, flow matching time $t$, amino acid identities $A \in \{1, \dots 20\}^L$, conditioning mask $\mathbf{m} \in \{0, 1\}^{T \times L \times (7K+14)}$
**Output:** flow velocity $v \in \mathbb{R}^{T \times L \times (7K+14)}$
$t \leftarrow \text{Embed}(t)$
**for** k $\leftarrow$ 1 to $K$ **do**
   $\mathbf{x}_k = \text{Embed}(A) + \sum_{k'} \text{Linear}(g_{i_k}^{-1} g_{i_{k'}})$
   **for** l $\leftarrow$ 1 to $num\_ipa\_layers$ **do**
      $\mathbf{x}_k = \text{InvariantPointAttentionLayer}(\mathbf{x}, g_k, t)$
   **end for**
**end for**
$\mathbf{x} = \sum_k \mathbf{x}_k + \text{Linear}(\boldsymbol{\chi}) + \text{Linear}(\boldsymbol{\chi}_{\text{cond}} \odot \mathbf{m}) + \text{Embed}(\mathbf{m})$
**for** l $\leftarrow$ 1 to $num\_transformer\_layers$ **do**
   $\mathbf{x} = \text{DiffusionTransformerAttentionLayer}(\mathbf{x}, t)$
**end for**
return DiffusionTransformerFinalLayer$(\mathbf{x}, t)$

---

**Algorithm 2** DiffusionTransformerAttentionLayer

---

**Input:** $\mathbf{x} \in \mathbb{R}^{T \times L \times C}$, time conditioning $t$
$(\alpha, \beta, \gamma)_{t, \ell, f} = \text{Linear}(t)$
$\mathbf{x} += g_\ell \odot \text{AttentionWithRoPE}(\gamma_\ell \odot \text{LayerNorm}(\mathbf{x}) + \beta_\ell, \dim = 1)$
$\mathbf{x} += g_t \odot \text{AttentionWithRoPE}(\gamma_t \odot \text{LayerNorm}(\mathbf{x}) + \beta_t, \dim = 0)$
$\mathbf{x} += g_m \odot \text{MLP}(\gamma_m \odot \text{LayerNorm}(\mathbf{x}) + \beta_m)$
return $\mathbf{x}$

---

---

**Algorithm 3** InvariantPointAttentionLayer

---

**Input:** $\mathbf{x} \in \mathbb{R}^{L \times C}$, time conditioning $t$, roto-translations $g \in SE(3)^L$

$(\alpha, \beta, \gamma)_{\ell, f} = \text{Linear}(t)$

$\mathbf{x} \mathrel{+}= \text{InvariantPointAttention}(\text{LayerNorm}(\mathbf{x}), g)$

$\mathbf{x} \mathrel{+}= g_\ell \odot \text{AttentionWithRoPE}(\gamma_\ell \odot \text{LayerNorm}(\mathbf{x}) + \beta_\ell)$

$\mathbf{x} \mathrel{+}= g_m \odot \text{MLP}(\gamma_m \odot \text{LayerNorm}(\mathbf{x}) + \beta_m)$

return $\mathbf{x}$

---

## B.2. Integrating Dirichlet Flow Matching

To additionally generate amino acid identities along with the trajectory dynamics, we integrate our SiT flow matching framework with Dirichlet flow matching (Stark et al., 2024). Specifically, we now parameterize a velocity network $v_\theta : \left(\mathbb{R}^{7K} \oplus \mathbb{R}^{20}\right)^{T \times L} \times [0, 1] \to \left(\mathbb{R}^{7K} \oplus \mathbb{R}^{20}\right)^{T \times L}$. No architecture modifications are necessary other than augmenting the tokens with one-hot tokens of residue identity, broadcasted across time. At training time, we sample from the Dirichlet probability path (rather than the Gaussian path) for those token elements. However, the parameterization is subtle as Dirichlet FM trains with cross-entropy loss, contrary to the standard flow-matching MSE loss. Thus, during *training time* we minimize the loss

$$\mathcal{L} = \mathbb{E}\left[\|v_\theta[\ldots, :-20] - u_t(\chi_t \mid \chi_1)\|^2 + \text{CrossEntropy}(\text{Softmax}(v_\theta[\ldots, -20:]), A)\right] \tag{2}$$

That is, we interpret the last 20 outputs in the channel dimension as logits over the 20 residue types. At *inference time*, on the other hand, we convert these logits to the Dirichlet FM flow field:

$$v_\theta' = \text{Concat}\left(v_\theta[\ldots, :-20], \sum_i \text{Softmax}(v_\theta[\ldots, -20:])_i \cdot u_{\text{DFM}}(\cdot \mid x_1 = i)\right) \tag{3}$$

where $u_{\text{DFM}}$ is the appropriate Dirichlet vector field from Stark et al. (2024).

## B.3. Conditional Generation

We control the conditional generation settings by simply setting appropriate entries of the conditioning mask $\mathbf{m}$ in Algorithm 1 to 1 or 0. Specifically,

- For the forward simulation setting, $\mathbf{m}[t, \ell, c] = \begin{cases} 1 & t = 1 \\ 0 & t \neq 1 \end{cases}$

- For the inpainting setting, $\mathbf{m}[t, \ell, c] = \begin{cases} 1 & t \in \{1, T\} \\ 0 & t \notin \{1, T\} \end{cases}$

- For the upsampling setting, $\mathbf{m}[t, \ell, c] = \begin{cases} 1 & t \,\%\, M = 1 \\ 0 & t \,\%\, M \neq 1 \end{cases}$ where $M$ is the upsampling factor.

- For the inpainting setting, $\mathbf{m}[t, \ell, c] = \begin{cases} 1 & \ell \in \mathcal{S}_{\text{known}} \\ 0 & \ell \notin \mathcal{S}_{\text{known}} \end{cases}$ where $\mathcal{S}_{\text{known}}$ is the set of residues in the known part of the trajectory.

We use 1 indexing to be consistent with the main text. In practice, in the inpainting setting we also mask out all torsion angles and withhold the amino acid identities for all residues. Further, we do not train the model to generate the torsions as all, such that the tokenization yields $\chi \in \mathbb{R}^{T \times L \times 7K}$. These interventions were observed to be necessary to prevent overfitting.

## C. Experimental Details

### C.1. Markov State Models

A Markov State Model (MSM) is a representation of a system's dynamics discretized into $r$ states $s \in \{1 \ldots r\}$ and a discrete timesteps separated by *time lag* $\tau$ such that the dynamics are approximately Markovian (Husic & Pande, 2018; Chodera & Noé, 2014; Pande et al., 2010). An MSM is parameterized with a vector $\boldsymbol{\pi}$ that assigns each state a stationary probability and a matrix $T$ containing the probabilities for transitioning from state $s_t$ to $s_{t+1}$ after one timestep, i.e., $T_{i,j} = p(s_{t+1} = j \mid s_t = i)$.

To build a Markov state model, we use PyEMMA (Scherer et al., 2015; Wehmeyer et al.) and its accompanying tutorials. Briefly, we first featurize molecular trajectories with all torsion angles as points on the unit circle, obtaining a $2m$-dimensional invariant trajectory where $m$ is the number of torsion angles. We run TICA on these trajectories with kinetic scaling and then run $k$-means clustering with $k = 100$ over the first few (5–10 chosen by PyEMMA) TICA coordinates. We then estimate an MSM over these 100 states and use PCCA+ spectral clustering (Röblitz & Weber, 2013) to further group these into 10 metastable states. Our final MSM is built from the discrete trajectory over these 10 metastable states. In all cases we use timelag $\tau = 100$ ps.

**Unconditionally sampling an MSM.** To unconditionally sample a trajectory of length $N$ from an MSM, we first sample the start state from the stationary distribution, i.e., $s_1 \sim \boldsymbol{\pi}$. We then iteratively sample each subsequent state as $s_{t+1} \sim T_{s_t,:}$.

**Sampling an MSM conditioned on a start state.** To sample a trajectory of length $N$ conditioned on a starting state $s_1$, we iteratively sample each subsequent state as $s_{t+1} \sim T_{s_t,:}$.

**Sampling an MSM conditioned on a start and end state.** For our transition path sampling evaluations in Section 3.2, we employ replica transition paths sampled from an MSM by conditioning on a start state $s_1$ and end state $s_N$. To do so, we iteratively sample each state between the conditioning states by utilizing the probability

$$p(s_{t+1} = j \mid s_t = i, s_N = k) = \frac{p(s_N = k \mid s_{t+1} = j, s_t = i)p(s_{t+1} = j \mid s_t = i)}{p(s_N = k \mid s_t = i)}. \tag{4}$$

Firstly, the term $p(s_{t+1} = j \mid s_t = i)$ is available in out transition matrix as $T_{i,j}$. Secondly, we obtain $p(s_N = k \mid s_t = i)$ as an entry of the $(N - t)th$ matrix exponential of the transition matrix. Specifically $p(s_N = k \mid s_t = i) = T_{i,k}^{(N-t)}$ where the superscript denotes a matrix exponential. Lastly, we obtain the term $p(s_N = k \mid s_{t+1} = j, s_t = i)$ by realizing that under the Markov assumption $p(s_N = k \mid s_{t+1} = j, s_t = i) = p(s_N = k \mid s_{t+1} = j)$. Further, $p(s_N = k \mid s_{t+1} = j) = T_{j,k}^{(N-t)-1}$.

Replacing the terms in Equation 4 results in

$$p(s_{t+1} = j \mid s_t = i, s_N = k) = \frac{T_{j,k}^{(N-t-1)}T_{i,j}}{T_{i,k}^{(N-t)}}. \tag{5}$$

Thus, we sample states $s_2 \ldots s_{N-1}$ iteratively as

$$s_{t+1} \sim \frac{T_{:,s_N}^{(N-t-1)}T_{s_t,:}}{T_{s_t,s_N}^{(N-t)}}. \tag{6}$$

### C.2. Tetrapeptide Molecular Dynamics

We run all-atom molecular dynamics simulations in OpenMM (Eastman et al., 2017) using the `amber14` force field parameters with `gbn2` implicit solvent or `tip3pfb` water model. Initial structures are generated with PyMOL, prepared with `pdbfixer`, and protonated at neutral pH. For explicit solvent, we prepare a solvent box with 10 Å padding and neutralize the system with sodium or chloride ions. All simulations are integrated with Langevin thermostat at 350K with hydrogen bond constraints, timestep 2 fs, and friction coefficient $0.3\,\mathrm{ps}^{-1}$ (explicit) or $0.1\,\mathrm{ps}^{-1}$ (implicit). For explicit solvent, nonbonded interactions are cut off at 10 Å with long-range particle-mesh Ewald. We first minimize the energy with L-BFGS and then equilibrate the system in the NVT ensemble for 20 ps. We then run 100 ns of production simulation in the NVT ensemble (implicit) or NPT ensemble with Monte Carlo barostat at 1 bar (explicit). We write heavy atom positions every 100 fs.

For explicit-solvent settings (forward simulation, interpolation, inpainting), we run simulations for 3109 training, 100 validation, and 100 test peptides. For implicit-solvent settings (upsampling), we run simulations for 2646 training, 100 validation, and 100 test peptides. All peptides are randomly chosen and split. Additionally, 5195 training and 100 validation implicit solvent simulations are run for the pentapeptide `MDGEN`.

### C.3. Evaluation Details

**Trajectory Featurization**    We featurize trajectories by selecting the sine and cosine of all torsion angles as the collective variables. Specifically, we featurize $\psi, \phi$ backbone angles and all $\chi$ sidechain torsion angles for each peptide. We then reduce dimensionality with Time-lagged Independent Components Analysis (TICA) (Pérez-Hernández et al., 2013) in PyEMMA (Scherer et al., 2015).

**Jensen-Shannon Divergence**    We compute the JSD as implemented in `scipy`, i.e.,

$$\sqrt{\frac{D(p \mid m) + D(q \mid m)}{2}} \tag{7}$$

where $m = (p + q)/2$. For the 1-dimenional JSD over torsion angles, we discretize the range $[-\pi, \pi]$ into 100 bins. For the 1-dimensional JSD over TIC-0, we discretize the range spanning the maximum and minimum values into 100 bins. For the 2-dimensional JSD over TIC-0,1 we discretize the space into $50 \times 50$ bins.

**Autocorrelation**    The autocorrelation of torsion angle $\theta$ at time lag $\Delta t$ is defined as $\langle \cos(\theta_t - \theta_{t+\Delta t}) \rangle$, corresponding to the inner product of $\theta_t, \theta_{t+\Delta t}$ on the unit circle. To compute the *decorrelation time* of a torsion angle, we subtract the baseline inner product $\langle \cos \theta \rangle^2 + \langle \sin \theta \rangle^2$, this is analogous to removing the mean of a real-valued time series before computing the autocorrelation. The decorrelation time is then defined as the time required for the autocorrelation to fall below $1/e$ of its initial value (which is always unity), with the subtracted baseline computed from the reference trajectory. In a small number of cases (21 torsions), the MDGEN trajectory did not decorrelate within 1000 frames (10 ns), and we exclude the angle from Figure 2F.

To compute the decorrelation time for TIC-0, we now define the autocorrelation as

$$\mathbb{E}[(y_t - \mu)(y_{t+\Delta t} - \mu)]/\sigma^2 \tag{8}$$

where $\mu, \sigma$ are computed from the *reference* trajectory. Hence, when computed for a sampled MDGEN trajectory, the autocorrelation may not start at unity and may not decay to zero. We report a *decorrelation time* if starts above and falls below 0.5 within 1000 frames (10 ns), which happens in 74 out of 100 cases as shown in Figure 2E.

**Interpolation**    In our interpolation or transition path sampling experiments, we sample 1000 trajectories of length 1ns for each of our 100 test tetrapeptides. We first select a start state $s_1$ and an end state $s_N$ that exhibits non-trivial transitions. To do so, we consider a reference MD simulation of 100 ns for the tetrapeptide and obtain an MSM as described in Appendix C.1. From the MSM's transition matrix $T$ and stationary distribution $\boldsymbol{\pi}$, we compute the flux matrix $F = T \odot Pi$ where $Pi$ is the square matrix with $\boldsymbol{\pi}$ in each column. The chosen start and end state is the row and column index of the smallest non-zero entry in $F$.

With the start state $s_1$ and end state $s_N$ selected, we sample 1000 start frames $\mathbf{x}_1$ and end frames $\mathbf{x}_N$ from the states. The 1000 start frames are sampled from all frames in the reference MD simulation that belong to state $s_1$. Analogously, the end frames are sampled from all frames belonging to state $s_N$. Using the 1000 pairs of start and end frames, we condition MDGEN on them and generate trajectories of 100 frames (1 ns). For evaluation, we discretize these trajectories under the 10-state clustering determined by the MSM of the reference MD simulation as described in Appendix C.1. Note that with the MSM lag time of 100 ps, these discrete trajectories are of length 10.

MD baselines. To sample transition paths of 1 ns between our selected start and end states, we employ MSMs built from replica MD simulations of varying lengths. For instance, for a replica MD simulation of 100ns, we first discretize its trajectory with the cluster assignments of the reference MD simulation (the same cluster assignments as we use to discretize the MDGEN ensemble and that we use for evaluation). Next, we estimate an MSM from the discretized trajectory. We then proceed to sample 1000 transition paths from the MSM as described in C.1 where the path is conditioned on an end and start state. In the event that the replica MSM has zero transition probability for transitioning out of the start state or zero

probability for transitioning into the end state (this occurs if the replica MD simulation never visited the start or end state), we treat all 1000 paths of the replica MD as having zero probability for our evaluation metrics which are further detailed in the following.

Computing TPS metrics. As described above, we obtain ensembles of 1000 discretized 1ns paths of 100 frames for both MDGEN and the replica MD simulations. For these, in Figure 3, we show a JSD, the rate of valid paths, and the average path probability. These metrics are computed with respect to the MSM of the reference MD simulation of length 100 ns.

- To compute the JSD, we draw 1000 discrete transition paths from the reference MD simulation and compute the probability of visiting each state from the frequency with which each state is visited. We do the same for the transition path ensemble of MDGEN (or the baseline) and compute the JSD between the categorical distributions as described above.

- The average path probability for an ensemble is the average of its paths' probabilities for transitioning from the start to the end state under the reference MSM. This probability can be computed as described in Appendix C.1.

- The valid path rate is the fraction of paths that have a non-zero probability.

**Inpainting**   In our inpainting experiments, we set out to design tetrapeptides that transition between two states. Considering the residue indices $1, 2, 3$ and $4$, we call the residues $1, 4$ the flanking residues which we condition on and $2, 3$ the inner residues which we aim to design. Specifically, we condition MDGEN on the trajectory of the flanking residues' backbone coordinates and generate the residue identities of the inner two residues. To carry out this design for a single tetrapeptide, we draw 1000 samples from MDGEN to estimate the mode of its joint distribution over the inner two residues.

The conditioning information (the trajectories of the outer two residues' backbone coordinates) is different for the two evaluation settings of designing transitions with *high flux* or for designing arbitrary transitions. However, for both of them, the start and end frames are provided as conditioning information via the key frames. In the high flux setting, the conditioning information is obtained by sampling 1000 paths from the reference MD simulation of length 10 ps with 100 frames that start and end in the desired states. These states are determined as those with the maximum flux between them (see the paths about interpolation above for a description of flux). When designing residues that give rise to arbitrary random paths, the trajectories are randomly sampled from the reference simulation.

After sampling 1000 pairs of residues for the inner two residues, we select the most frequently occurring pair as the final design. For this design, we report the sequence recovery (the fraction of residues that match the original sequence of the MD simulations from which the conditioning information was sampled).

Inpainting Baselines. We aim to assess the benefit that is obtained by the trajectory-based inference of MDGEN over a baseline that only takes the start frame or the start and end frame as input for designing residues that transition between two states. Thus, we construct DYNMPNN and S-MPNN. These baselines use the same architecture as MDGEN in the inpainting setting, but DYNMPNN only obtains the start and end frames as key frames and via their roto-translation offsets for the first and last frames. S-MPNN is the analog with only the first frame.

Notably, in the inpainting setting, **MDGen** and the baselines do not treat torsion angles, and all torsion angle entries of the SE(3)-invariant tokens are set to 0. Furthermore, the model does not take the amino acids of the flanking residues as input. We make this choice of withholding all information about amino acid identities since otherwise, the models overfit on the arbitrary identities of the flanking residues and do not generalize to the test set.

**Protein Simulations**   For training and evaluation on proteins, we use trajectories from the ATLAS dataset (Vander Meersche et al., 2024), which includes 3 replicates of 100 ns explicit-solvent, all-atom simulations for each of 1390 non-membrane protein monomers. The proteins are chosen from the PDB as representatives of all available ECOD domains and are thus structurally non-redundant. We split the dataset into 1265 training, 39 validation, and 82 test proteins by PDB release date following Jing et al. (2024). At training time, we randomly select a protein, select one of the three replicates, subsample every 40 frames, obtaining a training target with 250 frames. We train with random crops of up to 256 residues, but draw samples for the full protein at inference time. To compute statistical similarity of the MDGEN ensembles with the ground truth MD ensembles, we compare the 250 frames with 30k pooled frames from all three trajectories. Baseline metrics and runtimes for AlphaFlow and MSA subsampling are taken directly from Jing et al. (2024). Analysis and visualization code for Table 4 and Figure 6 are provided courtesy of Jing et al. (2024).

**Runtime**   MD runtimes in Table 1 are tabulated on a NVIDIA T4 GPU. All MDGEN experiments are carried out on NVIDIA A6000 GPUs. AlphaFlow and MSA subsampling runtimes in Table 4 are tabulated on NVIDIA A100 GPUs by Jing et al. (2024).

*Table 3.* Sequence recovery for the inner two peptides when conditioning on the partial trajectory (MDGEN), the two terminal frames (DynMPNN), or a single frame (S-MPNN).

| Method | High Flux | Random Path |
|--------|-----------|-------------|
| MDGen | **52.1%** | **62.0%** |
| DynMPNN | 17.4% | 24.5% |
| S-MPNN | 16.3% | 13.5% |

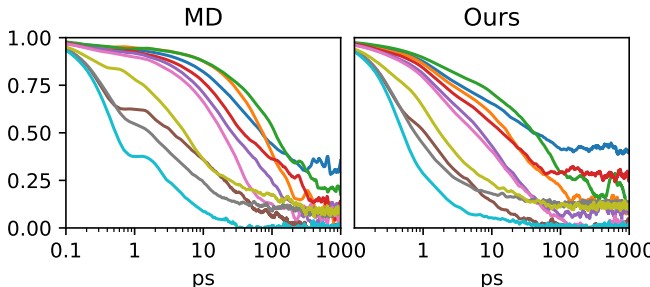

*Figure 5.* Autocorrelation functions of `MDGEN` sidechain torsion angles computed from a single MD trajectory (*left*) versus a **single model sample** with Hyena (*right*), capturing dynamical timescales spanning four orders of magnitude.

## D. Additional Results

### D.1. Additional Tasks

**Inpainting Design.** We aim to sample trajectories conditioned on the dynamics of the two flanking residues of the tetrapeptide; in particular, the model determines the identities and dynamics of the two inner residues. We focus on *dynamics scaffolding* as one possible higher-level objective of inpainting: given the conformational transition of the observed residues, we hope to design peptides that support flux between the corresponding Markov states.

Thus, for each test peptide, we select a 100-ps transition between the two most well-connected Markov states, mask out the inner residue identities and dynamics, and inpaint them with our model. To evaluate the designs, we compute the fraction of generated residue types that are identical to the tetrapeptide in which the target transition is known to occur. We compare MDGEN with a bespoke inverse folding baseline that is provided the two terminal states (i.e., two fully observed MD frames), and thus designs peptides that support the two modes (rather than additionally a partially-observed *transition* between them). We call this baseline DYNMPNN, and it otherwise has the same architecture and settings as MDGEN. We find (Table 3) that MDGEN recovers the ground-truth peptide substantially more often than DynMPNN when conditioned on a high-flux path or (as a sanity check) a random path from the reference simulatiom.

**Scaling to Long Trajectories.** Although Section 3.1 showed that our model can emulate long trajectories, this was limited to rollouts of 1000 frames at a time with coarse 10 ps timesteps, potentially missing faster dynamics or disrupting slower dynamics. Thus, we investigate generating extremely long consistent trajectories that capture timescales spanning several orders of magnitude *within a single model sample*. To do so, we replace the time attention in our baseline SiT architecture with a non-causal Hyena operator (Poli et al., 2023), which has $O(N \log N)$ rather than $O(N^2)$ overhead. We overfit on 100k-frame, 10-ns trajectories of the pentapeptide `MDGEN` and compare the torsional autocorrelation functions computed from a *single* generated trajectory with a *single* ground truth trajectory (Figure 5). Although not yet comparable to the main set of forward simulation experiments due to data availability and architectural expressivity reasons, these results demonstrate proof-of-concept for longer context lengths in future work.

**Protein Simulation.** To demonstrate the applicability of our method for larger systems such as proteins, we train a model to emulate explicit-solvent, all-atom simulations of proteins from the ATLAS dataset (Vander Meersche et al., 2024) conditioned on the first frame (i.e., forward simulation). We follow the same splits as Jing et al. (2024). Due to the much larger number of residues, we generate samples with 250 frames and 400 ps timestep, such that a single sample emulates the 100 ns ATLAS reference trajectory. The difficulty of running fully equilibrated trajectories for proteins prevents the

*Table 4.* Median results on test protein ensembles ($n = 82$). Runtimes are reported per sample structure or frame.

|  | MDGEN | AlphaFlow | MSA sub. |
|---|---|---|---|
| Pairwise RMSD $r$ ↑ | 0.48 | 0.48 | 0.22 |
| Global RMSF $r$ ↑ | 0.50 | 0.60 | 0.29 |
| Per-target RMSF $r$ ↑ | 0.71 | 0.85 | 0.55 |
| Root mean $\mathcal{W}_2$ dist. ↓ | 2.69 | 2.61 | 3.62 |
| MD PCA $\mathcal{W}_2$ dist. ↓ | 1.89 | 1.52 | 1.88 |
| % PC-sim $\geq 0.5$ ↑ | 10 | 44 | 21 |
| Weak contacts $J$ ↑ | 0.51 | 0.62 | 0.40 |
| Exposed residue $J$ ↑ | 0.29 | 0.41 | 0.27 |
| Runtime (s) | 0.2 | 70 | 4 |

MD                                          Ours

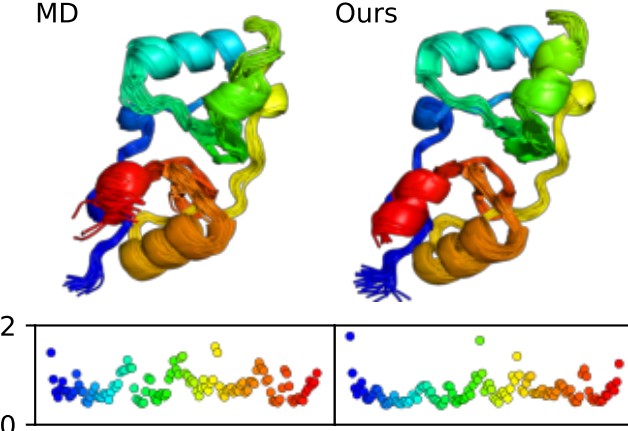

*Figure 6.* MD vs generated ensembles for `6uof_A`, with C$\alpha$ RMSFs plotted by residue index (Pearson $r = 0.74$).

construction of Markov state models used in our main evaluations. Instead, we compare statistical properties of forward simulation ensembles following Jing et al. (2024). Our ensembles successfully emulate the ground-truth ensembles at a level of accuracy between AlphaFlow and MSA subsampling while being orders of magnitude faster per generated structure than either (Table 4).

## D.2. Forward Simulation

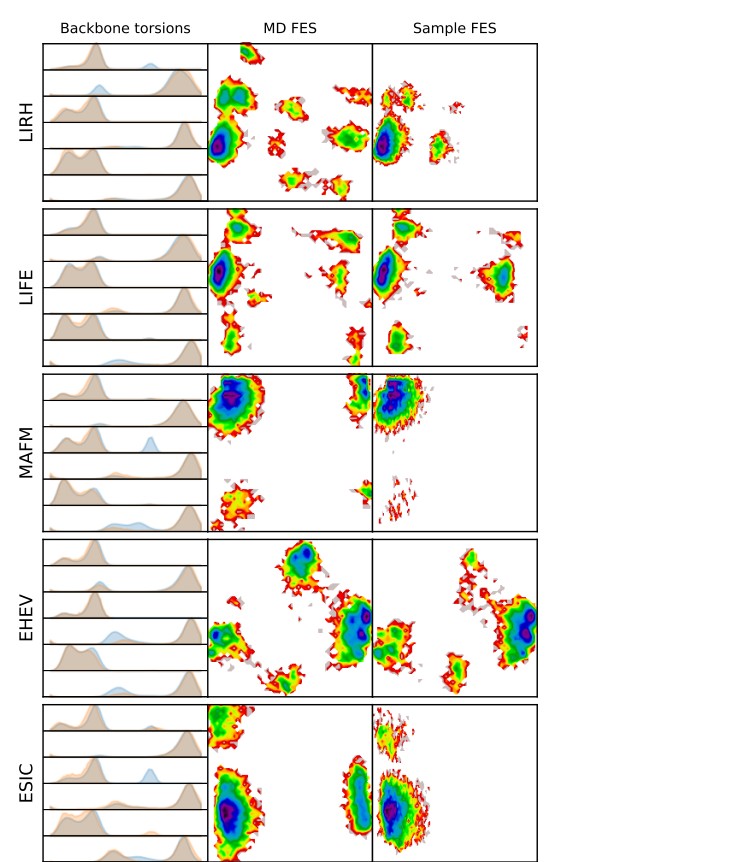
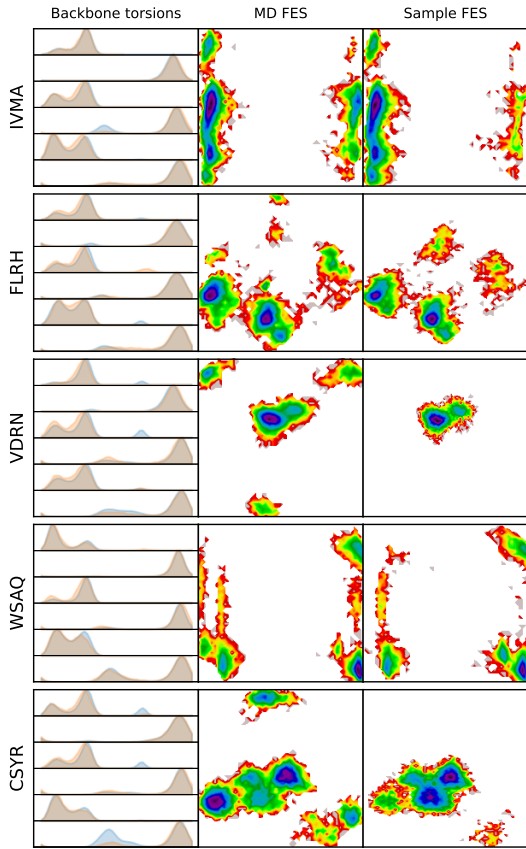

*Figure 7.* Additional backbone torsion angle distributions (orange from MD, blue from samples) and free energy surfaces along the top two TICA components for 10 randomly chosen test peptides.

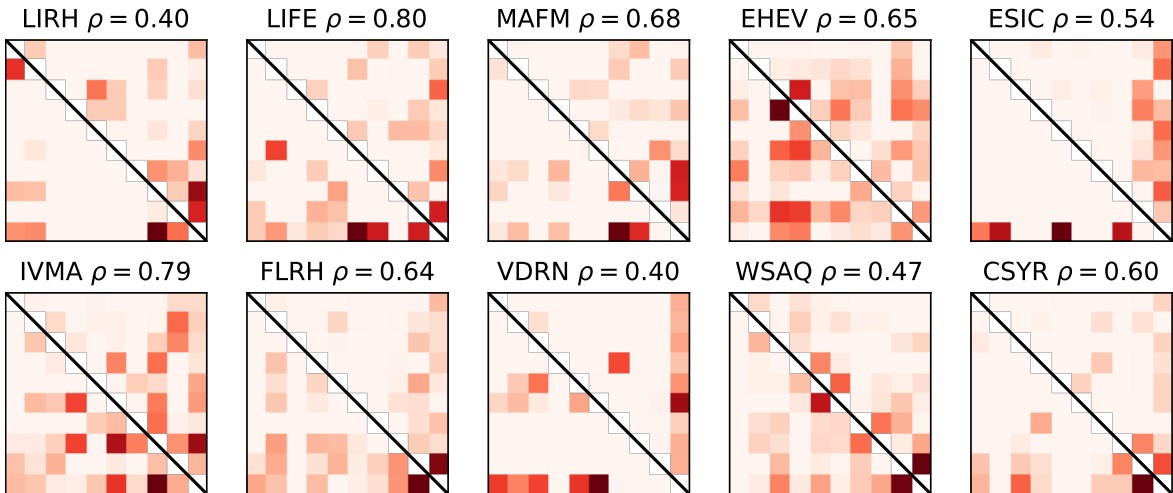

*Figure 8.* Flux matrices between MSM metastable states computed from reference MD trajectories (upper right) and MDGEN trajectories (bottom left) for 10 random test peptides (the matrices are symmetric). Cells are colored by the square root of the flux, with darker indicating high flux. The Spearman correlation between the entries is shown.

## D.3. Interpolation

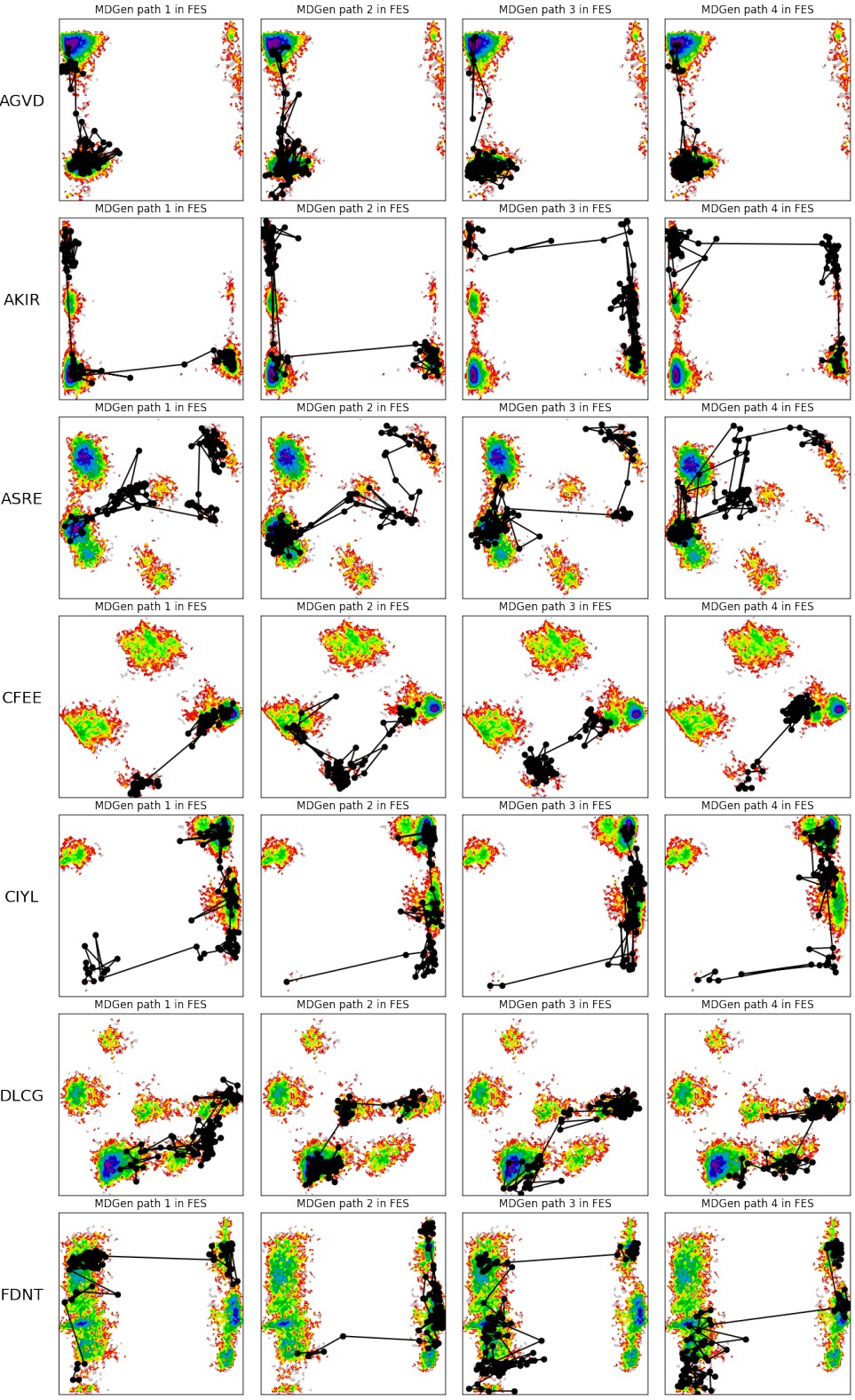

*Figure 9.* Four of 1000 transition paths of MDGEN for several tetrapeptides in the test set.

## D.4. Upsampling

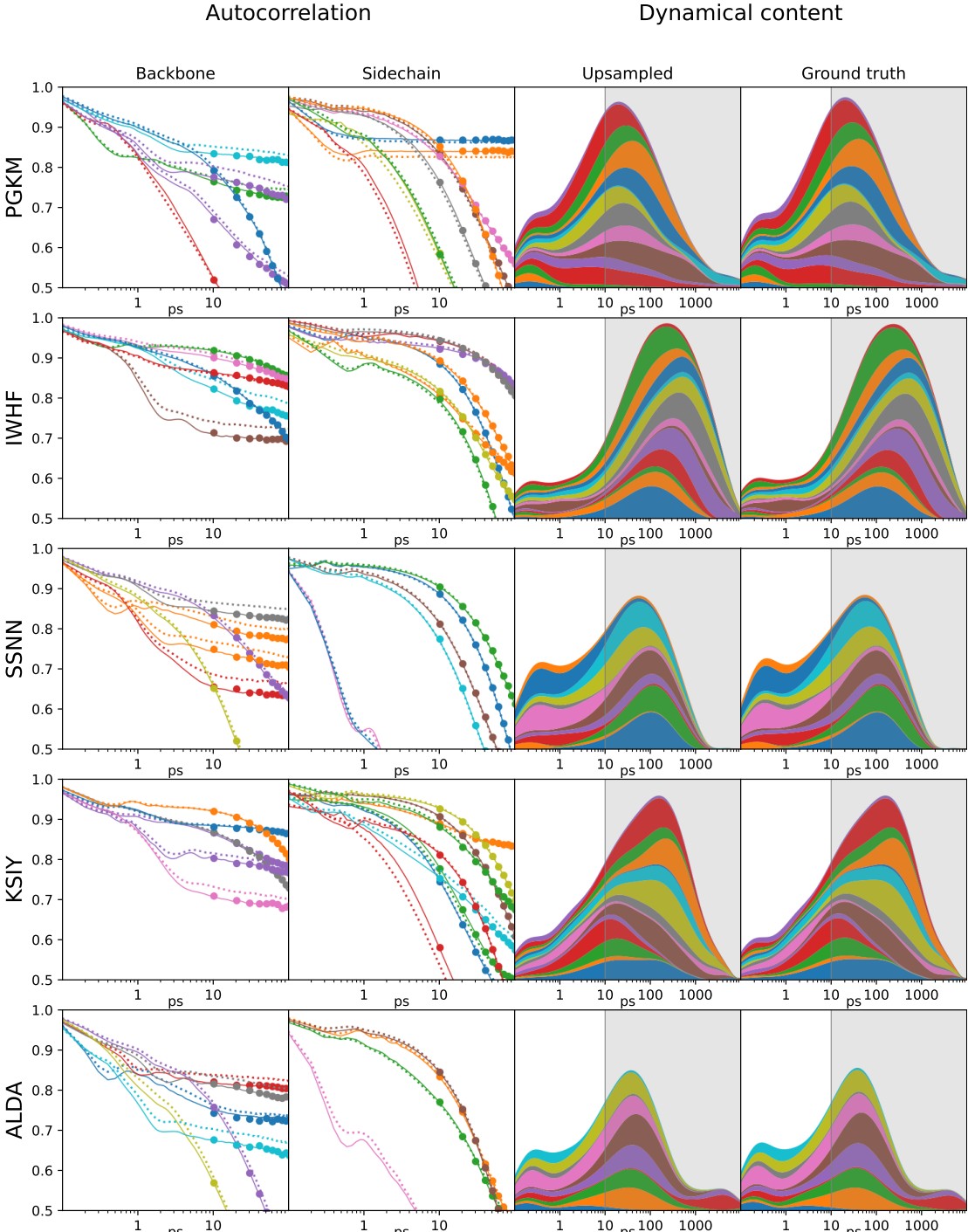

*Figure 10.* **Recovery of fast dynamics via trajectory upsampling** for random test peptides. (*Left*) Autocorrelations of each torsion angle from (—) the original 100 fs-timestep trajectory, (●) the subsampled 10 ns-timestep trajectory, and (⋅⋅⋅) the reconstructed 100 fs-timestep trajectory (all length 100 ns). (*Right*) Dynamical content as a function of timescale from the upsampled vs. ground truth trajectories, stacked for all torsion angles (same color scheme). The subsampled trajectory contains only the shaded region and our model recovers the unshaded region.

## D.5. Inpainting

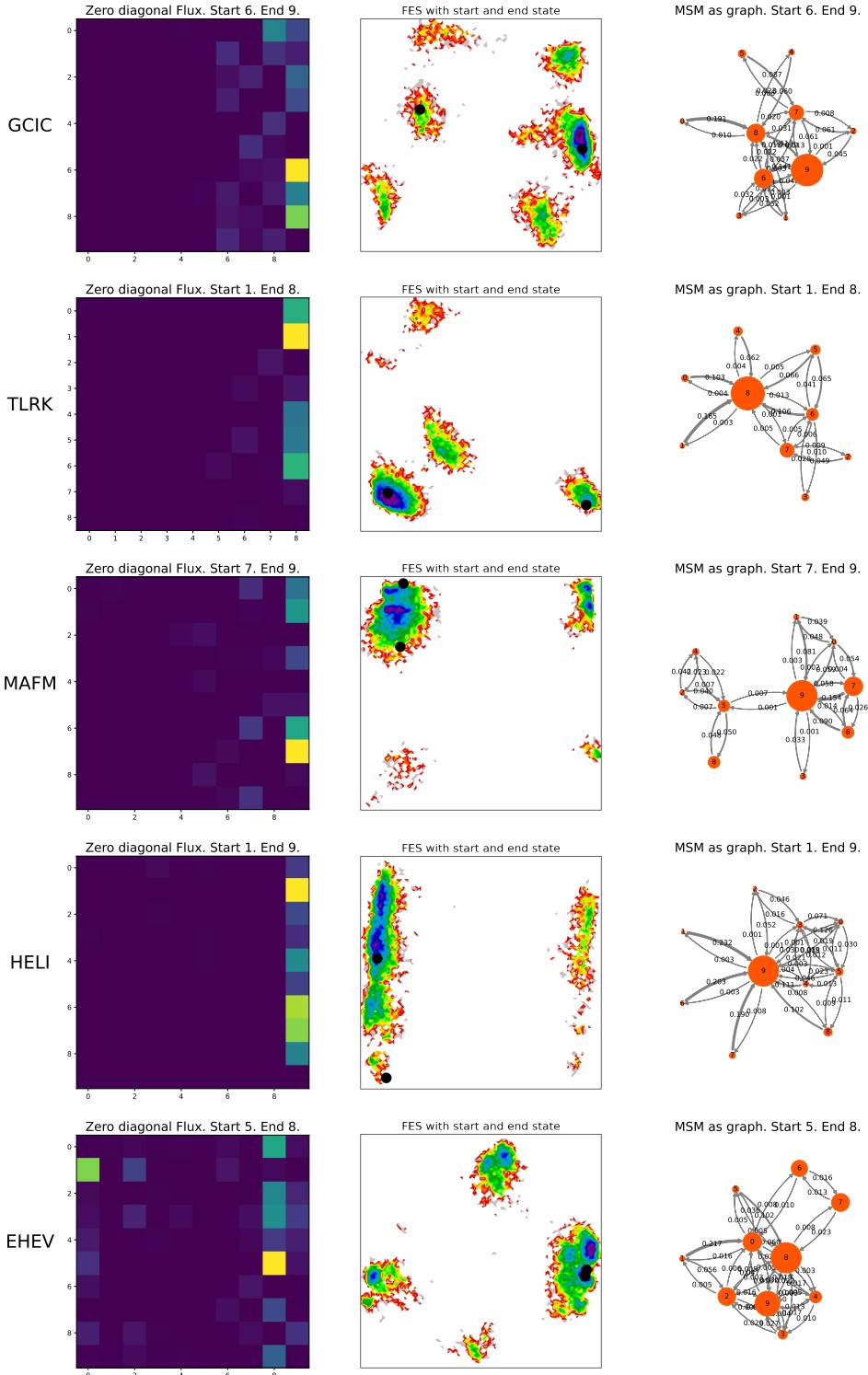

Figure 11. For six tetrapeptides, we show the states that we chose in our design experiments when designing transitions between the highest flux states. Column 1 shows the flux matrix with zeros on the diagonal. Column 2, the free energy surface of a 100 ns simulation and the selected start and end states based on the highest flux in the flux matrix. Column 3, the MSM that was built from the MD simulation.

