# OpenReview forum: "Generative Modeling of Molecular Dynamics Trajectories"
_ICML.cc/2024/Workshop/ML4LMS — ML4LMS Poster_

### Official Review · Reviewer_dGvn · 2024-06-10

**Rating:** 8
**Confidence:** 4

**Review:**

**Summary:**

The authors present a framework for the modeling of MD trajectories, able to simulate accelerated time evolution but also perform more advanced tasks such as path sampling.
The central idea relies on the modelling of the trajectory as a fixed-size time series, over which a generative process can be learnt. This is done in the general framework of stochastic interpolants and through a network containing several blocks taken from the literature (IPA, DiT).

**Pros:**

The approach is very interesting, explained quite clearly, and it tackles an important problem.

The results shown cover very different scenarios and are quite convincing.

**Cons:**

While there are a lot of benefits to this approach, it would be interesting to hear some of the possible limitations.
In particular, as the architecture is only described in the appendix, and in terms of existing high level blocks, it is not very clear to me how flexible the approach would be to changes in the system under study. For example: longer simulations can be obtained by chaining the models, but can it be extended to arbitrary sized sequences? (The supplementary material seems to imply this can be done at least in inference, but even for proteins L was kept fixed in training)

More in general, since the basis in this work is specialized for amino acids, it would be interesting to read some discussion over possible extensions to all atom systems.

Also, I understand this might not be possible, but it would be interesting to see a comparison to existing approaches (besides Table 4), at least for common tasks (e.g. Timewarp is also validated on tetrapeptides).

**Minor remarks:**

I could not understand what is shown in orange in Fig.2F: are these the sidechains?

---

### Official Review · Reviewer_bLgu · 2024-06-11
**The authors propose MDGen a novel method based on stochastic interpolants adaptations to generate MD trajectories**

**Rating:** 8
**Confidence:** 4

**Review:**

**Summary**: The authors explain their contribution clearly, effectively bridging molecular dynamics concepts with artificial intelligence. The method is soundly formulated and well adapted for the interpolants literature. The use of SE(3)-invariant tokenization of molecular trajectories is a well-chosen strategy to handle the rotational and translational invariances of molecular structures, contributing significantly to the model's robustness. The experiments demonstrate the potential of the method in relevant scenarios, although they are primarily limited to small-sized peptides (with additional examples in the Appendix). Comparisons to MD are fair and show the strengths of the method in capturing dynamic behaviors.

**Strengths** : MDGEN is demonstrated to be capable of multiple tasks—forward simulation, interpolation, upsampling, and inpainting—showing the flexibility and potential broad applicability of the approach. The model shows strong performance in generating realistic molecular trajectories, capturing free energy surfaces, dynamical content, and transition paths that align well with MD simulations. The model's ability to handle these tasks efficiently, with significant computational speedups, is noteworthy.

**Weaknesses**: The evaluations are primarily focused on short peptides and single-chain proteins. This work would greatly benefit from full scale experimentation and reporting on protein structures.

_Errata_: (1) There is a missing parenthesis in line page 148. (2) Probably a missing "contains" after "which" in line page 158 on the Experiments sections.

---

### Official Review · Reviewer_43TN · 2024-06-12
**Great paper on a relevant and challenging topic**

**Rating:** 9
**Confidence:** 4

**Review:**

The authors tackle the very difficult problem of modeling protein dynamics. The solutions applied here are novel and non-trivial. The work is very exciting, has high potential for impact, and deserves an easy acceptance to the workshop.

The paper is very dense and could be edited to fit the format better. Since each use case requires separate training, each could be a paper on its own. However, I understand not wanting to split it into too many pieces. I imagine that some parts ended up in the SI to accommodate the 5-page limit of the workshop. The discussion has been cut short for the same reason. I think that the workshop version of it could be limited to 1-2 tasks (e.g., forward simulation and interpolation); this would allow for clearer communication. The paper

A few additional comments:
1. Will the code and models be publicly available?
2. I would include MDGen in the title; it would be easier to remember.
3. Since tokenization is key for the paper, an expanded description of parametrization would be beneficial. It's unclear why the dimension of the token is 7K+14 and not 7K+7 (offsets + torsion angles of the residue)? How are small residues tokenized (the ones without 4 dihedrals in the side chain)?
4. Information about training resources and times would be great.
5. Table 1: "MDGen" label would work better than "Ours."
6. Table 1: Errors would be nice (bootstrap).
7. What is the sequence similarity between train and test tetrapeptides?
8. Section 3.1: "We report the Jensen... Figure 2 and Table 2." Do the authors mean Table 1?
9. Section 3.2: It would be interesting to see how the model performs at sampling transitions that are possible in the expected timescales. Does it correctly model transition paths? Connected to this: How does the model perform with training on larger intervals than 1 ns?
10. Section 3.3: How does the model perform on bigger timescales, 10 ps > 100 fs?
11. Section 3.3: "... for two test peptides..." there is only one test peptide in the figure.
12. Inpainting is lacking from the results section. It is in the SI, but the intro builds an expectation that it will be in the main text.
13. The title "Opportunities" in the discussion is not needed.

SI:
14. Additional description of how the tetrapeptide set was made would be great.
15. Section C.2: What do the authors mean by "hydrogen bond constraints"? Is it maybe hydrogen (not bond) constraints?
16. Section C.3: Inpainting: Does this setting mean we work only with backbone atoms? If the flanking residue identity is known, why are they not fully parameterized with their side chains? Is the objective here only to predict the identity of masked residues or also conformation? Some additional comments are needed.
17. Protein Simulations: Just a suggestion: I think "fast-folding" proteins would be a great example of a test set for "bigger" proteins. They have well-known dynamics and have been studied in multiple papers. The long reference MDs are also available.
* Lindorff-Larsen, Kresten, et al. "How fast-folding proteins fold." Science 334.6055 (2011): 517-520.
* Majewski, Maciej, et al. "Machine learning coarse-grained potentials of protein thermodynamics." Nature Communications 14.1 (2023): 5739.
18. Table 4: Data for reference MD should be included for comparison.